Gall-ID: tools for genotyping gall-causing phytopathogenic bacteria

Davis II Edward W. 1 2
Weisberg Alexandra J. 1
Tabima Javier F. 1
Grunwald Niklaus J. 1 2 3 4
Chang Jeff H. changj@science.oregonstate.edu 1 2 3
1 Department of Botany and Plant Pathology, Oregon State University , Corvallis , OR , United States
2 Molecular and Cellular Biology Program, Oregon State University , Corvallis , OR , United States
3 Center for Genome Research and Biocomputing, Oregon State University , Corvallis , OR , United States
4 Horticultural Crops Research Laboratory, USDA-ARS , Corvallis , OR , United States
Zhao Min
Electronic publication date: 2016 Jul 19
Publication date: 2016
Volume: 4
Electronic Location ID: e2222
Received 2016 Apr 29; Accepted 2016 Jun 15
Copyright: ©2016 Davis II et al.
Copyright year: 2016
Copyright holder: Davis II et al.
License: This is an open access article distributed under the terms of the Creative Commons Attribution License, which permits unrestricted use, distribution, reproduction and adaptation in any medium and for any purpose provided that it is properly attributed. For attribution, the original author(s), title, publication source (PeerJ) and either DOI or URL of the article must be cited.
License URL: https://creativecommons.org/licenses/by/4.0/

Keywords: Taxonomy, Genomes, Molecular diagnostics, Multilocus sequence analysis, Average nucleotide identity, Agrobacterium, Rhodococcus

Funding: National Institute of Food and Agriculture 2014-51181-22384 USDA Agricultural Research Service 5358-22000-039-00D USDA National Institute of Food and Agriculture 2011-68004-30154 USDA ARS Floriculture Nursery Research Initiative Oregon State University National Science Foundation Graduate Research Fellowship DGE-1314109 National Science Foundation #1340112 #1127112 This work was supported by the National Institute of Food and Agriculture, US Department of Agriculture award 2014-51181-22384 (JHC and NJG). Partial support also was provided by the USDA Agricultural Research Service Grant 5358-22000-039-00D (NJG), USDA National Institute of Food and Agriculture Grant 2011-68004-30154 (NJG), and the USDA ARS Floriculture Nursery Research Initiative (NJG). EWD is supported by a Provost’s Distinguished Graduate Fellowship awarded by Oregon State University. This material is based upon work supported by the National Science Foundation Graduate Research Fellowship under Grant No. DGE-1314109 to EWD. The summer camps were supported by National Science Foundation awards IOS #1340112 and #1127112 to Pankaj Jaiswal. The funders had no role in study design, data collection and analysis, decision to publish, or preparation of the manuscript.

==============================
Understanding the population structure and genetic diversity of plant pathogens, as well as the effect of agricultural practices on pathogen evolution, is important for disease management. Developments in molecular methods have contributed to increase the resolution for accurate pathogen identification, but those based on analysis of DNA sequences can be less straightforward to use. To address this, we developed Gall-ID, a web-based platform that uses DNA sequence information from 16S rDNA, multilocus sequence analysis and whole genome sequences to group disease-associated bacteria to their taxonomic units. Gall-ID was developed with a particular focus on gall-forming bacteria belonging to Agrobacterium, Pseudomonas savastanoi, Pantoea agglomerans, and Rhodococcus. Members of these groups of bacteria cause growth deformation of plants, and some are capable of infecting many species of field, orchard, and nursery crops. Gall-ID also enables the use of high-throughput sequencing reads to search for evidence for homologs of characterized virulence genes, and provides downloadable software pipelines for automating multilocus sequence analysis, analyzing genome sequences for average nucleotide identity, and constructing core genome phylogenies. Lastly, additional databases were included in Gall-ID to help determine the identity of other plant pathogenic bacteria that may be in microbial communities associated with galls or causative agents in other diseased tissues of plants. The URL for Gall-ID is http://gall-id.cgrb.oregonstate.edu/.

Introduction

Diagnostics

Determining the identity of the disease causing pathogen, establishing its source of introduction, and/or understanding the genetic diversity of pathogen populations are critical steps for containment and treatment of disease. Proven methods for identification have been developed based on discriminative phenotypic and genotypic characteristics, including presence of antigens, differences in metabolism, or fatty acid methyl esters, and assaying based on polymorphic nucleotide sequences (Alvarez, 2004). For the latter, polymerase chain reaction (PCR) amplification-based approaches for amplifying informative regions of the genome can be used. These regions should have broadly conserved sequences that can be targeted for amplification but the intervening sequences need to provide sufficient resolution to infer taxonomic grouping.

The 16S rDNA sequence is commonly used for identification (Stackebrandt & Goebel, 1994). Because of highly conserved regions in the gene sequence, a single pair of degenerate oligonucleotide primers can be used to amplify the gene from a diversity of bacteria, and allow for a kingdom-wide comparison. In general, the sequences of the amplified fragments have enough informative polymorphic sites to delineate genera, but do not typically allow for more refined taxonomic inferences at the sub-genus level (Janda & Abbott, 2007). Multilocus sequence analysis (MLSA) leverages the phylogenetic signal from four to ten genes to provide increased resolution, and can distinguish between species and sometimes sub-species (Wertz et al., 2003; Zeigler, 2003). MLSA however, is more restricted than the use of 16S rDNA sequences and may not allow for comparisons between members of different genera. MLSA also requires more time and effort to identify informative and taxon-specific genes as well as develop corresponding oligonucleotide primer sets.

Whole genome sequences can also be used. This is practical because of advances in next generation sequencing technologies. The key advantage of this approach is that availability of whole genome sequences obviates the dependency on a priori knowledge to provide clues on taxonomic association of a pathogen and the need to select and amplify marker genes. Also, whole genome sequences provide the greatest resolution in terms of phylogenetic signal, and sequences that violate assumptions of phylogenetic analyses (e.g., not vertically inherited) can be removed from studies to allow for robust analyses. Briefly, sequencing reads of genome(s) are compared to a high quality draft or finished reference genome sequence to identify variable positions between the genome sequences (Pearson et al., 2009). The positions core to the compared genome sequences are aligned and used to generate a phylogenetic tree. Alternatively, whole genome sequences can be used to determine average nucleotide identities (ANI) between any sufficiently similar pair, e.g., within the same taxonomic family, of genome sequences to determine genetic relatedness (Goris et al., 2007; Kim et al., 2014). ANI can be used to make taxonomic inferences, as a 95% threshold for ANI has been calibrated to those used to operationally define bacterial species based on 16S rDNA (>94% similarity) and DNA–DNA hybridization (DDH; 70%) (Goris et al., 2007). Finally, the whole genome sequences can be analyzed to inform on more than just the identity of the causative agent and provide insights into mechanisms and evolution of virulence. However, a non-trivial trade-off is that processing, storing, and analyzing whole genome sequence data sets require familiarity with methods in computational biology.

Agrobacterium spp.

Members from several taxa of Gram-negative bacteria are capable of causing abnormal growth of plants. Members of Agrobacterium are the most notorious causative agents of deformation of plant growth. These bacteria have been classified according to various schemes that differ in the phenotypic and genetic characteristics that were used. Its taxonomic classification has been a subject of multiple studies (Young et al., 2001; Farrand, Berkum & Oger, 2003; Young, 2003). Here, we will use the classification scheme that is based on disease phenotype and more commonly encountered in the literature. Within Agrobacterium there are four recognized groups of gall-causing bacteria. A. tumefaciens (also known as Rhizobium radiobacter (Young et al., 2001), formerly A. radiobacter, genomovar G8 forms A. fabrum (Lassalle et al., 2011)) can cause crown gall disease that typically manifests as tumors on roots or stem tissue (Gloyer, 1934; Kado, 2014). A. tumefaciens can infect a wide variety of hosts and the galls can restrict plant growth and in some cases kill the plant (Gloyer, 1934; Schroth et al., 1988). Other gall causing clades include A. vitis (restricted to infection of grapevine), A. rubi (Rubus galls), and A. larrymoorei, which is sufficiently different based on results of DNA–DNA hybridization studies to justify a species designation (Hildebrand, 1940; Ophel & Kerr, 1990; Bouzar & Jones, 2001). The genus was also traditionally recognized to include hairy-root inducing bacteria belonging to A. rhizogenes, as well as non-pathogenic biocontrol isolates belonging to A. radiobacter (Young et al., 2001; Velázquez et al., 2010). Members of Agrobacterium are atypical in having multipartite genomes which, in some cases, include a linear replicon (Allardet-Servent et al., 1993).

A Ti (tumor-inducing) plasmid imparts upon members of Agrobacterium the ability to genetically modify its host and cause dysregulation of host phytohormone levels and induce gall formation (Sachs, 1975). The Ti plasmid contains a region of DNA (T-DNA) that is transferred and integrated into the genome of the host cell (Van Larebeke et al., 1974; Chilton et al., 1977; Thompson et al., 1988; Ward et al., 1988; Broothaerts et al., 2005). Conjugation is mediated by a type IV secretion system, encoded by genes located outside the borders of the T-DNA on the Ti plasmid (Thompson et al., 1988). Within the T-DNA are key genes that encode for auxin, cytokinin, and opine biosynthesis (Morris, 1986; Binns & Costantino, 1998). Expression of the former two genes in the plant leads to an increase in plant hormone levels to cause growth deformation whereas the latter set of genes encode enzymes for the synthesis of modified sugars that only organisms with the corresponding opine catabolism genes can use as an energy source (Bomhoff et al., 1976; Montoya et al., 1977). The latter genes are located outside the T-DNA borders on the Ti plasmid (Zhu et al., 2000).

Pseudomonas savastanoi

Pseudomonas savastanoi (formerly Pseudomonas syringae pv. savastanoi, (Gardan et al., 1992)) is the causal agent of olive knot disease, typically forming as aerial tumors on stems and branches. Phytopathogenicity of P. savastanoi is dependent on the hrp/hrc genes located in a pathogenicity island on the chromosome (Sisto, Cipriani & Morea, 2004). These genes encode for a type III secretion system, a molecular syringe that injects type III effector proteins into host cells that collectively function to dampen host immunity (Chang, Desveaux & Creason, 2014). Phytopathogenicity of P. savastanoi is also associated with the production of phytohormones. Indole-3-acetic acid may have an indirect role as a bacterial signaling molecule (Aragon et al., 2014). A cytokinin biosynthesis gene has also been identified on plasmids in P. savastanoi and strains cured of the plasmid caused smaller galls but were not affected in growth within the galls (Iacobellis et al., 1994; Bardaji et al., 2011).

Pantoea agglomerans

Pantoea agglomerans (formerly Erwinia herbicola) is a member of the Enterobacteriaceae family. P. agglomerans can induce the formation of galls on diverse species of plants (Cooksey, 1986; Burr et al., 1991; Opgenorth, Hendson & Clark, 1994; DeYoung, Copeman & Hunt, 1998; Vasanthakumar & McManus, 2004). Phytopathogenicity is dependent on the pPATH plasmid (Manulis & Barash, 2003; Weinthal et al., 2007). This plasmid contains a pathogenicity island consisting of an hrp/hrc cluster and operons encoding for the biosynthesis of cytokinins, indole-3-acetic acid, and type III effectors (Clark et al., 1993; Lichter et al., 1995; Nizan et al., 1997; Mor et al., 2001; Nizan-Koren et al., 2003; Barash & Manulis, 2005; Barash et al., 2005; Barash & Manulis-Sasson, 2007). As is the case with P. savastanoi, mutants of the hrp/hrc genes abolish pathogenicity whereas mutations in the phytohormone biosynthesis genes led to galls of reduced size (Manulis et al., 1998; Mor et al., 2001; Nizan-Koren et al., 2003; Barash & Manulis-Sasson, 2007).

Rhodococcus spp.

Members within the Gram-positive Rhodococcus genus can cause leafy gall disease to over 100 species of plants (Putnam & Miller, 2007). The phytopathogenic members of this genus belong to at least two genetically distinct groups of bacteria, with R. fascians (formerly Corynebacterium fascians) being the original recognized species (Goodfellow, 1984; Creason et al., 2014a). It is suggested that R. fascians upsets levels of phytohormones of the plant to induce gall formation. However, unlike Ti plasmid-carrying Agrobacterium, it is hypothesized that R. fascians directly synthesizes and secretes the cytokinin phytohormone (Stes et al., 2013). Phytopathogenicity is most often associated with a linear plasmid, which carries a cluster of virulence loci, att, fasR, and fas (Creason et al., 2014b). The functions for the translated products of att are unknown but the sequences have homology to proteins involved in amino acid and antibiotic biosynthesis (Maes et al., 2001). The fasR gene is necessary for full virulence; the gene encodes a putative transcriptional regulator (Temmerman et al., 2000). Some of the genes within the fas operon are necessary for virulence, as many of the fas genes encode proteins with demonstrable functions in cytokinin metabolism (Crespi et al., 1992). In rare cases, the virulence loci, or variants therein, are located on the chromosome (Creason et al., 2014b).

We developed Gall-ID to aid in determining the genetic identity of gall-causing members of Agrobacterium, Pseudomonas, Pantoea, and Rhodococcus. Users can provide sequences from 16S rDNA or gene sets used in MLSA, and Gall-ID will automatically query curated databases and generate phylogenetic trees to group the query isolate of interest and provide estimates of relatedness to previously characterized species and/or genotypes. Users can also submit short reads from whole genome sequencing projects to query curated databases to search for evidence for known virulence genes of these gall-causing bacteria. Finally, users can download tools that automate the analysis of whole genome sequencing data to infer genetic relatedness based on MLSA, average nucleotide identity (ANI), or single nucleotide polymorphisms (SNPs).

Material and Methods

Website framework and bioinformatics tools

The Gall-ID website and corresponding R shiny server backend are based on the Microbe-ID platform (Tabima et al., 2016) but include major additions and modifications: Auto MLSA, Auto ANI, BLAST with MAFFT, and the WGS Pipeline. The MLSA framework website was extended to support building Neighbor-Joining trees using incomplete distance matrices (NJ∗) using the function njs() in the R package PHYLOCH (Paradis, Claude & Strimmer, 2004). The MLSA framework was also modified to use the multiple sequence alignment program MAFFT using the R package PHYLOCH (Katoh & Toh, 2008; Heibl, 2013; Katoh & Standley, 2013). This allows user-submitted sequences to be added to pre-existing sequence alignments using the MAFFT “–add” function, to dramatically reduce the computational time required for analysis.

The server hosting the Gall-ID tools is running Centos Linux release 6.6, MAFFT version 7.221, SRST2 version 0.1.5, Bowtie 2 version 2.2.3, and Samtools version 0.1.18. Gall-ID uses R version 3.1.2 with the following R packages: Poppr version 1.1.0.99 (Kamvar, Tabima & Grünwald, 2014), Ape version 3.1-1 (Paradis, Claude & Strimmer, 2004), PHYLOCH version 1.5-5, and Shiny version 0.8.0.

The Auto MLSA tool was developed previously (Creason et al., 2014a). Briefly, Auto MLSA does the following: BLAST (either TBLASTN or BLASTN) to query NCBI user-selectable databases and/or local databases and retrieve sequences, filter out incomplete sets of gene sequences, align gene sequence individually, concatenate aligned gene sequences, determine the best substitution model (for amino acid sequences), filter out identical sequences, append key information to sequences, and generate a partition file for RAxML (Stamatakis, 2014). Auto MLSA also has the option of using Gblocks to trim alignments (Castresana, 2000). Auto MLSA was modified to use the NCBI E-utilities, implemented in BioPerl, to associate accession numbers with taxon IDs, species names, and assembly IDs (Stajich, 2002). For organisms without taxon identifiers, Auto MLSA will attempt to extract meaningful genus and species information from the NCBI nucleotide entry. Gene sequences are linked together using assembly IDs, which allows for genomes with multiple chromosomes to be compared, without having to rely on potentially ambiguous organism names. When assembly IDs are unavailable, whole genome sequences are linked using the four letter WGS codes, and, as a last resort, sequences will be associated using their nucleotide accession number. The disadvantage of using the latter approach is that organisms with multiple replicons, each with its own accession number, will be excluded from analysis. Auto MLSA is available for download from the Gall-ID website. Detailed instructions for using the tools are provided.

The Auto ANI script automates the calculations of ANI for all pairwise combinations for any number of input genome sequences. Each of the supplied genome sequences are chunked into 1,020 nt fragments and used as queries in all possible reciprocal pairwise BLAST searches. Parameters for genome chunk size, percent identity, and percent coverage have default values set according to published guidelines but can be changed by the user (Goris et al., 2007; Creason et al., 2014a). BLAST version 2.2.31+ was used with recommended settings, as previously described in Creason et al. (2014a): –task blastn –dust no –xdrop_gap 150 –penalty −1 –reward 1 –gapopen 5 –gapextend 2 (Goris et al., 2007). BLAST hits above the user-specified cut-offs (30% identity, 70% coverage, by default) are averaged to calculate the pairwise ANI values.

BLAST+ version 2.2.27 has been tested and works, but this version is currently unsupported. Versions 2.2.28–2.2.30 of BLAST+ have an undocumented bug that prevents efficient filtering using -max_hsps and -max_target_seqs and precludes their use in ANI calculation. Hence BLAST 2.2.31+ is the preferred and recommended version.

Sequences downloaded from NCBI are linked using assembly IDs. All accession types from NCBI are supported, assuming accession numbers are provided in the header line of the FASTA file. Locally generated genome sequences are also supported, in FASTA format, provided they follow the specified header format listed in the user guide. Alternatively, an auxiliary script is provided to rename headers within user-generated FASTA files to the supported format.

The WGS Pipeline was written in bash shell script and Perl. Paired Illumina sequencing reads located in the “reads” folder of the pipeline are processed in pairs. The program SMALT (Ponstingl, 2013) is used to align reads to a reference genome and produce CIGAR format output files (Ponstingl, 2013). The SSAHA_pileup program converts the CIGAR format files into individual pileup files (Ning, Cox & Mullikin, 2001). The pileup output is then combined with any additional supplied pre-computed pileup files and used to produce a core alignment of sites shared by 90% of the represented isolates. The optional “remove_recombination.sh” script runs the program Gubbins (Croucher et al., 2014) to remove sites identified as potentially acquired by recombination. Finally, the program RAxML is used to produce a maximum-likelihood phylogenetic tree with non-parametric bootstrap support (Stamatakis, 2014). By default 20 maximum likelihood tree searches are performed, and the “autoMRE” criterion is used to determine the number of non-parametric bootstrap replicates.

Table 1 Statistics for the WGS Pipeline.

WGS Pipeline step	Statistic	Value	
generate_pileup.sh (1 cpu)	Number of input paired read sets	19	
Average runtime per pileup (hh:mm:ss)	00:42:01	
Total runtime (hh:mm:ss)	13:18:14	
generate_core_alignment.sh (1 cpu)	Total pileup alignment length	5,947,114 bp	
90%-shared core alignment length	855,355 bp	
Total runtime (hh:mm:ss)	00:15:32	
remove_recombination.sh (10 cpus)	Number of core polymorphic sites	177,961 bp	
core SNP alignment length (w/o putative recombinant SNPs)	174,819 bp	
Computational time (hh:mm:ss)	04:25:32	
Actual runtime (hh:mm:ss)	00:29:28	
Figure output runtime (hh:mm:ss)	00:13:03	
generate_phylogeny.sh (raxmlHPC-PTHREADS-AVX, 10 cpus)	Time to optimize RAxML parameters (hh:mm:ss)	00:02:32	
Time to compute 20 ML searches (hh:mm:ss)	00:34:53	
Number of bootstrap replicates (RAxML autoMRE)	50	
Time to compute 50 bootstrap searches (hh:mm:ss)	01:02:09	
Total runtime (hh:mm:ss)	01:39:34	
All	Total runtime (hh:mm:ss)	15:55:51	

The WGS Pipeline test analysis was performed and benchmarked using 10 cores of a cluster server running Centos Linux release 6.6 and containing four AMD Opteron™ 6376 2.3 Ghz processors (64 cores total) and 512 GB of RAM (Table 1). The versions of the tools used in tests of this pipeline were Perl version 5.10.1, SMALT version 0.7.6, SSAHA_pileup version 0.6, Gubbins version 1.1.2, and RAxML version 8.1.17. The default parameters for WGS Pipeline were used (20 ML search trees, “autoMRE” cutoff for bootstrap replicates) with the exception that the maximum-allowed percentage gaps in the Gubbins recombination analysis was increased to 50% in order to retain strain D188. The WGS Pipeline scripts were also modified to not ask for user input on the command line in order to run in a Sun Grid Engine (SGE) cluster environment.

Vir-Search uses the program SRST2, which employs Bowtie 2 and Samtools, with the “–gene_db” function to align the reads to custom databases of the virulence genes (Li et al., 2009; Inouye et al., 2012; Langmead & Salzberg, 2012; Inouye et al., 2014). The identity of the virulence genes that the reads input by the user align to the read coverage and depth, and the name of the strain corresponding to the most similar allele are parsed from the SRST2 output and reported to the user as a static webpage. Users are emailed a link to results once the analysis is complete. The submitted sequencing reads are deleted from the server immediately after completion, and results are available only to those with a direct link to the results webpage.

Datasets

The 16S and MLSA gene sequences were downloaded from the genome sequences of the following reference strains: Agrobacterium strain C58, Rhodococcus strain A44a, P. savastanoi pv. phaseolicola 1448A, and P. agglomerans strain LMAE-2, C. michiganensis subsp. nebraskensis NCPPB 2581, D. dadantii strain 3937, P. atrosepticum strain 21A, R. solanacearum strain GMI1000, X. oryzae pv. oryzicola strain CFBP2286, and X. fastidiosa subsp. fastidiosa GB514 (NCBI assembly ID: GCF_000092025.1, GCF_000760735.1, GCF_000012205.1, GCF_000814075.1, GCF_000355695.1, GCF_000147055.1, GCF_000740965.1, GCF_000009125.1, GCF_001042735.1, and GCF_000148405.1, respectively). The gene sequences were used as input for the Auto MLSA tool in BLAST searches carried out against complete genome sequences in the NCBI non-redundant (nr) and whole genome sequence (wgs) databases. The Auto MLSA parameters were: minimum query coverage of 50% (90% for the 16S plant pathogen dataset) and e-value cutoffs of 1e–5 for nr and 1e–50 for wgs. BLAST searches were limited to the genus for the bacteria of interest, with the exceptions of Agrobacterium, which was limited to Rhizobiaceae, and P. savastanoi, which was limited to the P. syringae group. BLAST searches were completed in August of 2015. The Auto MLSA tool uses MAFFT aligner to produce multiple sequence alignments for each gene (Katoh & Standley, 2013). The Gblocks trimmed alignment output of Auto MLSA was not used because Gall-ID aligns user-submitted gene sequences to each full gene alignment (Castresana, 2000).

Table 2 Strain identity of 14 isolates associated with crown gall.

Isolate name	Host	Positive ID based on	# high quality read pairs	Clade (based on 16S rDNA)	# of virulence genes ID’ed	
13-2099-1-2	Quaking Aspen	virD2 PCR	1,244,074	Agrobacterium	63	
13-626	Pear	virD2 PCR	220,903	Agrobacterium	2 (nocM, nocP)	
AC27/96	Pieris	Not pathogenic	826,690	Rhizobium	1 (tssD)	
AC44/96	Pieris	No reaction to hybridization probes	1,404,002	Rhizobium	0	
B131/95	Peach/Almond Rootstock	Pathogenicity assay	539,283	Agrobacterium	46	
B133/95	Peach/Almond Rootstock	Pathogenicity assay	1,199,902	Agrobacterium	46	
B140/95	Peach/Almond Rootstock	Response to 20 different biochemical and physiological tests	448,314	A. tumefaciens	51	
N2/73	Cranberry gall	Response to 20 different biochemical and physiological tests	1,345,404	A. tumefaciens	64	
W2/73	Euonymus	Response to 20 different biochemical and physiological tests	1,244,159	A. rubi	51	
15-1187-1-2a	Yarrow	virD2 PCR	508,223	A. tumefaciens	39	
15-1187-1-2b	Yarrow	virD2 PCR	299,970	A. tumefaciens	38	
14-2641	Rose	No data	698,756	Serratia	0	
15-172	Leucanthemum	Colony morphology on selective media	384,308	A. tumefaciens	56	
15-174	Leucanthemum	Colony morphology on selective media	753,570	A. tumefaciens	58	

Bacterial strains, growth conditions, nucleic acid extraction, and genome sequencing

Strains of Agrobacterium were grown overnight in Lysogeny Broth (LB) media at 28 °C, with shaking at 250 rpm (Table 2). Cells were pelleted by centrifugation and total genomic DNA was extracted using a DNeasy Blood and Tissue kit (Qiagen, Venlo, Netherlands). DNA was quantified using a QuBit Fluorometer (Thermo Fisher, Eugene, Oregon) and libraries were prepared using an Illumina Nextera XT DNA Library Prep kit, according to the instructions of the manufacturer, with the exception that libraries were normalized based on measurements from an Agilent 2100 Bioanalyzer (Agilent Technologies, Santa Clara, CA). Each library was assigned an individual barcode using an Illumina Nextera XT Index kit. Libraries were multiplexed and sequenced on an Illumina MiSeq to generate 300 bp paired-end reads. Sequencing was done in the Center for Genome Research and Biocomputing Core Facility (Oregon State University, Corvallis, OR). Sickle was used to trim reads based on quality (minimum quality score cutoff of 25, minimum read length 150 bp after trimming) (Joshi & Fass, 2011). Read quality was assessed prior to and after trimming using FastQC (FastQC, Cambridge, UK). Paired reads for each library were de novo assembled using Velvet version 1.2.10 with the short paired read input option (“-shortPaired”), estimated expected coverage (“-exp_cov auto”), and default settings for other parameters (Zerbino & Birney, 2008). Genome sequences were assembled using a range of input hash lengths (k-mer sizes), and the final assembly for each isolate was identified based on those with the best metrics for the following parameters: total assembly length (5.0∼7.0 Mb), number of contigs, and N50. Paired reads for each library were error corrected and assembled using SPAdes versions 3.6.2 and 3.7.0, with the careful option (“–careful”) and kmers 21, 33, 55, 77, and 99. Scaffolds shorter than 500 bp and with coverage less than 5X were removed from the SPAdes assemblies prior to analysis.

Results and Discussion

Gall-ID (http://gall-id.cgrb.oregonstate.edu/) is based on the Microbe-ID platform and uses molecular data to determine the identity of plant pathogenic bacteria (Tabima et al., 2016). Gall-ID is organized into modules shown as tabs that allow users to choose from one of four options for analyzing data (Fig. 1).

Figure 1 Overview of Gall-ID diagnostic tools.

DNA sequence information can be used to reveal the identity of the causative agent (unknown isolate) of disease. Tools associated with “Gall Isolate Typing” and “Phytopath-type” use 16S rDNA or pathogen-specific MLSA gene sequences to infer the identity of the isolate by comparing the sequences to manually curated sequence databases. Tools associated with “Whole Genome Analysis” and “Vir-Search” use Illumina short sequencing reads to characterize pathogenic isolates. The former tab provides downloadable tools to infer genetic relatedness based on SNPs (WGS Pipeline) or average nucleotide identity (Auto ANI). The “Vir-Search” tab provides an on-line tool to quickly map short reads against a database of sequences of virulence genes.

Gall Isolate Typing

The “Gall Isolate Typing” tab provides online tools to use molecular data, either 16S or sequences of marker genes used for MLSA, to group isolates of interest into corresponding taxonomic units that include gall-causing pathogens. Users must first select the appropriate taxonomic group, Agrobacterium, Pseudomonas, Pantoea, or Rhodococcus for comparison. For some of these taxonomic groups, multiple gene sets used in MLSA are available, and the user must therefore select the appropriate set for analysis. FASTA formatted gene sequences are input, and after selecting the appropriate options for alignment and tree parameters, a phylogenetic tree that includes the isolate of interest is generated and displayed. The tree parameters include choice of distance matrix, tree generating algorithm (Neighbor-Joining or UPGMA), and number of bootstrap replicates. A sub-clade of the tree containing only the isolate of interest and its nearest sister taxa is displayed to the right of the full tree. The tree can be saved as a Newick file or as a PDF. An example sequence from Agrobacterium can be loaded by clicking the “Demo” button located in the Agro-type tab.

Phytopath-Type

The “Phytopath-Type” tab provides online tools for the analysis of other non-gall-causing pathogens important in agriculture (Mansfield et al., 2012). This tool is similar in function to the “Gall Isolate Typing” tools, except it is not limited to a single taxon of pathogen. A database of 16S rDNA sequences from genera of important bacterial phytopathogens (Pseudomonas syringae group, Ralstonia, Agrobacterium, Rhodococcus, Xanthomonas, Pantoea, Xylella, Dickeya, Pectobacterium, and Clavibacter) is available for associating a bacterial pathogen to its genus. Additionally, for Clavibacter, Dickeya, Pectobacterium, Ralstonia, Xanthomonas, and Xylella, the user can use MLSA to genotype isolates of interest. As is the case with Gall Isolate Typing, a phylogenetic tree will be generated and displayed, associating the isolate of interest to the most closely related genotype in the curated databases.

Vir-Search

The “Vir-Search” tab provides an online tool for using user-input read sequences of a genome to search for the presence of homologs of known virulence genes. Users select a taxonomic group (Agrobacterium, P. savastanoi, P. agglomerans, or Rhodococcus) to designate the set of virulence genes to search against. Users also determine a minimum percent gene coverage and maximum allowed percent identity divergence, and upload single or paired read files in FASTQ format. The user-supplied read sequences are then aligned to the chosen virulence gene dataset on the Gall-ID server. Once the search is complete, a link to the final results is sent to a user-provided email address. Results display the percent coverage of the virulence genes and the percent similarity of the covered sequences. If the query identifies multiple alleles of virulence genes from different sequenced strains, the Vir-Search tool will report the strain name associated with the best-mapped allele. User-submitted data and results are confidential and submitted sequencing reads are deleted from the Gall-ID server upon completion of the analysis.

Table 3 Manually curated datasets developed for Gall-ID.

Database	Bacterial group	# of isolates used in Gall-ID	References	
“Agro-type” tool (Agrobacterium)	
MLSA (dnaK, glnA, gyrB, recA, rpoB, thrA, truA)	Rhizobiaceae	199	Pérez-Yépez et al. (2014)	
MLSA (atpD, gapA, gyrB, recA, rplB)	Rhizobiaceae	188	Alexandre et al. (2008)	
dnaJ	Rhizobiaceae	198	Alexandre et al. (2008)	
16S rDNA	Rhizobiaceae	245		
“Rhodo-type” tool (Rhodococcus)	
MLSA (ftsY, infB, rpoB, rsmA, secY, tsaD, ychF)	Rhodococcus	85	Adékambi et al. (2011)	
16S rDNA	Rhodococcus	66		
“Panto-type” tool (Pantoea agglomerans)	
MLSA (fusA, gyrB, leuS, pyrG, rplB, rpoB)	Pantoea, Erwinia	356	Delétoile et al. (2009)	
16S rDNA	Pantoea, Erwinia	352		
“Pseudo-type” tool (Pseudomonas savastanoi)	
MLSA (gapA, gltA, gyrB, rpoD)	Pseudomonas syringae	158	Hwang et al. (2005)	
MLSA (acnB, fruK, gapA, gltA, gyrB, pgi, rpoD)	Pseudomonas syringae	153	Sarkar & Guttman (2004)	
16S rDNA	Pseudomonas syringae	161		
“Phytopath-type” tool	
16S rDNA	Rhodococcus, Agrobacterium, Pseudomonas syringae, Ralstonia, Xanthomonas, Pantoea, Erwinia, Xylella, Dickeya, Pectobacterium, Clavibacter, Rathayibacter	345		
MLSA (atpD, dnaK, gyrB, ppK, recA, rpoB)	Clavibacter	7	Jacques et al. (2012)	
MLSA (dnaA, gyrB, kdpA, ligA, sdhA)	Clavibacter	7	Tancos, Lange & Smart (2015)	
MLSA (dnaA, dnaJ, dnaX, gyrB, recN)	Dickeya	40	Marrero et al. (2013)	
MLSA (acnA, gapA, icdA, mdh, pgi)	Pectobacterium	54	Kim et al. (2009)	
MLSA (adk, egl, fliC, gapA, gdhA, gyrB, hrpB, ppsA)	Ralstonia	28	Castillo & Greenberg (2007)	
MLSA (dnaK, fyuA, gyrB, rpoD)	Xanthomonas	348	Young et al. (2008)	
MLSA (acvB, copB, cvaC, fimA, gaa, pglA, pilA, rpfF, xadA)	Xylella	17	Parker, Havird & De La Fuente (2012)	

Databases of Gall-ID

A key component of the tools associated with the aforementioned tabs is the manually curated databases of gene sequences. The literature was reviewed to identify validated taxon-specific sets of genes for MLSA of taxa with gall-causing bacteria as well as other pathogens that affect agriculture (Table 3) (Sarkar & Guttman, 2004; Hwang et al., 2005; Castillo & Greenberg, 2007; Alexandre et al., 2008; Young et al., 2008; Delétoile et al., 2009; Kim et al., 2009; Adékambi et al., 2011; Jacques et al., 2012; Parker, Havird & De La Fuente, 2012; Marrero et al., 2013; Pérez-Yépez et al., 2014; Tancos, Lange & Smart, 2015). The sequences for the corresponding genes were subsequently extracted from the whole genome sequences of reference strains. Auto MLSA was employed to use the gene sequences as queries in BLAST searches. Auto MLSA is based on a previously developed set of Perl scripts to automate retrieving, filtering, aligning, concatenating, determining best substitution models, appending key identifiers to sequences, and generating files for tree construction (Creason et al., 2014a). Gene sets in which there were less than 50% query sequence coverage for all of the genes were excluded to ensure that the databases contained only taxonomically informative sequences. Each gene set database was manually checked for duplicate strains, large gaps in gene sequences, poor sequence alignment, and mis-annotated taxonomic information. Each of the MLSA databases used in the Gall-ID tools is also available for download on the “Database Downloads” tab of the Gall-ID website. The 16S rDNA databases were populated in a similar manner, with one small exception. For the Phytopath-Type tool, the sequence of the 16S rRNA-encoding gene from C58 of Agrobacterium was used as a query to retrieve corresponding sequences from 345 isolates distributed across the different genera of plant pathogenic bacteria.

To populate the database for virulence genes, the literature was reviewed to identify genes with demonstrably necessary functions for the pathogenicity of Agrobacterium spp., R. fascians, P. savastanoi, and P. agglomerans (Thompson et al., 1988; Ward et al., 1988; Lichter et al., 1995; Nizan et al., 1997; Manulis et al., 1998; Zhu et al., 2000; Maes et al., 2001; Vereecke et al., 2002; Nizan-Koren et al., 2003; Sisto, Cipriani & Morea, 2004; Nissan et al., 2006; Barash & Manulis-Sasson, 2007; Matas et al., 2012). The gene sequences were downloaded from corresponding type strains in the NCBI nucleotide (nr) database or from nucleotide sequences in the NCBI nr database. The downloaded virulence gene sequences were then used as input for the Auto MLSA tool to retrieve sequenced alleles from other isolates of the same taxa. The downloaded alleles were manually inspected to ensure only pathogenic strains were represented. The database was formatted for SRST2.

Whole Genome Analysis

The analyses of whole genome sequence datasets can be computationally intensive, which is prohibitive for online tools. Therefore, the “Whole Genome Analysis” tab provides downloadable software pipeline tools for users to employ their institutional infrastructure or a cloud computing service to analyze whole genome sequencing reads (Illumina HiSeq or MiSeq). There are two options in this tab, the first, “WGS Pipeline: Core Genome Analysis,” provides a download link and instructions for using the WGS Pipeline tool to generate a phylogeny based on the core genome sequence or core set of single nucleotide polymorphisms (SNPs). The second option, “Auto ANI: Average Nucleotide Identity Analysis,” provides a download link for the Auto ANI tool and detailed instructions for its use in calculating all possible pairwise ANI within a set of genome sequences.

The WGS Pipeline is a set of scripts that automates the use of sequences from Illumina-based paired reads derived from whole genome sequencing projects to determine core genome sequences or core SNPs and generate phylogenetic trees (Fig. 2). This pipeline uses SMALT and SSAHA2 pile-up pipeline to align sequencing reads to an indexed reference genome sequence and generate a pileup file, respectively (Ning, Cox & Mullikin, 2001; Ponstingl, 2013). The WGS Pipeline then combines the pileup files along with other pre-computed pileup files to derive a core genome alignment defined based on regions that are shared between at least 90% of the compared genome sequences. Users have the option of using Gubbins to remove regions that are flagged as potentially derived from recombination (Croucher et al., 2014). Invoking Gubbins will also remove all non-polymorphic sites from the alignments, thus yielding a SNP alignment that is based only on polymorphic sites that are identified as vertically inherited and shared between at least 90% of the compared genome sequences. Finally, the user can use either the core genome sequence or core SNP alignment and RAxML to generate a maximum likelihood (ML) phylogeny (Stamatakis, 2014).

Figure 2 Flowchart for the WGS Pipeline.

Scripts and the programs that each script runs are boxed and presented along the left. The logic flow of the WGS Pipeline tool is presented along the right. Rectangles with rounded corners, inputs and outputs; boxes outlined in red, processes. The inputs, outputs, and processes are matched to the corresponding script and program.

Users must place their input files in the correspondingly named folders in order to run the WGS pipeline. The pipeline down weights reads with a Q score of <30, requires a minimum depth of 12 and relies on a minimum threshold of 75% for consensus base calling. Users concerned with sequencing quality may, prior to running the WGS pipeline, run programs such as FastQC, Trimmomatic, Sickle, and/or BBDuk to filter reads based on quality threshold (FastQC, Cambridge, UK; Joshi & Fass, 2011; Bolger, Lohse & Usadel, 2014; BBMap, at http://sourceforge.net/projects/bbmap/). Paired read sequences for each genome are read from the “reads” folder, while a SMALT index named as “reference” and placed in the “index” folder will be used as a reference to align to. Identifiers are taken from the prefix of the read pair file names and used to name the output pileup files and taxa in the phylogeny. The read pair file names must have the suffixes “.1.fastq” and “.2.fastq” for files with forward and reverse read sequences, respectively. The read sequences must be in FASTQ format and because of requirements of the SMALT program, each paired read name must end in “.p1k” and “.q1k” for forward and reverse reads, respectively. If the input read sequences are not in the proper format, the user may run the included optional script “prepare_for_pileup.sh” to format read names. If the user has pre-computed SMALT pileup files prepared using the same SMALT index, the files may be placed in the “pileup” folder and will also be included in the analysis. The user may be prompted to input the length of the inserts for each sequencing library. Users also have the option of changing the number of ML searches or non-parametric bootstrap replicates when building a phylogeny (default values are 20 ML searches, autoMRE cutoff criterion for bootstrap replicates).

Pre-built SMALT indices for reference genome sequences from strain C58 of Agrobaterium and strain A44a of Rhodococcus, as well as pre-computed pileup files for 17 publicly available Rhodococcus genome sequences, are available for download on the Gall-ID website. Detailed usage instructions and download links for the pipeline scripts are located in the “WGS Pipeline: Core Genome Analysis” tool in the “Whole Genome Analysis” tab of Gall-ID.

Previously developed scripts for ANI analysis were rewritten and named Auto ANI. The current version of these scripts alleviates dependencies on our institutional computational infrastructure and increases the scalability of analyses (Creason et al., 2014a). Results are saved in a manner that enables analyzing additional genomes without having to re-compute ANI values for previously calculated comparisons. All BLAST searches are done in a modular manner and can be modified to run on a computer cluster with a queuing system such as the Sun Grid Engine. There are no inherent restrictions on the numbers of pairwise comparisons that can be performed. The data are output as a tab delimited matrix of all pairwise comparisons and can also easily be sorted and resorted based on any reference within the output. Additionally, genome sequences with evidence for poor quality assemblies can be easily filtered out. A distance dendrogram based on ANI divergence can also be generated; a python script is available for download (Chan et al., 2012; Creason et al., 2014a).

Validation of tools available in Gall-ID

We validated the efficacy of the online tools available from the Gall Isolate Typing, and Vir-Search tabs. DNA from 14 isolates were prepared, barcoded, and sequenced on an Illumina MiSeq (Table 2). Of these isolates, the identities of 11 were previously verified as Agrobacterium. The remaining three were associated with plant tissues showing symptoms of crown gall disease but were not tested or had results inconsistent with being a pathogenic member of the Agrobacterium genus. The reads were trimmed for quality and first de novo assembled within each library using the Velvet assembler (Zerbino & Birney, 2008). The 16S gene sequences were identified and extracted from the assemblies and used as input for the Agro-type tool. The 16S gene sequences from each of the 11 isolates originally typed as Agrobacterium clustered accordingly; isolate 13-2099-1-2 is shown as an example (Fig. 3A). The 16S sequence from isolates AC27/96, AC44/96, and 14-2641 were more distant from the 16S sequences of Agrobacterium (Table 2). The isolates AC27/96 and AC44/96 grouped more closely with various Rhizobium species, while subsequent analysis using the Phytopath-Type tool suggested isolate 14-2641 was more closely related to members of Erwinia, Dickeya, and Pectobacterium (Table 2, Fig. S1). A search against the NCBI nr database revealed similarities to members of Serratia.

Figure 3 Validation of the Agro-type and Vir-Search tools.

(A) An unrooted Neighbor Joining phylogenetic tree based on 16s rDNA sequences from Agrobacterium spp. The 16S rDNA sequence was identified and extracted from the genome assembly of Agrobacterium isolate 13-2099-1-2 and analyzed using the tool available in the Agro-type tab. The isolate is labeled in red, as “query_isolate”; inset shows the clade that circumscribes the isolate. (B) Screenshot of output results from Vir-Search. Paired 2 × 300 bp MiSeq short reads from Agrobacterium isolate 13-2099-1-2 were analyzed using the Vir-Search tool in Gall-ID. Reference virulence gene sequences that were aligned are indicated with a green plus (“+”) icon and the lengths and depths of the read coverage are reported (must exceed user-specified cutoffs, which were designated as 90% minimum coverage and 20% maximum sequence divergence). Virulence genes that failed to exceed user-specific cutoffs for read alignment parameters are indicated with a red “X”. Virulence genes are grouped into categories based on their function in virulence.

The trimmed read sequences were used as input for the Vir-Search tool as an additional step to confirm the identity of these isolates. Paired read sequences for each of the 14 isolates were individually uploaded to the Gall-ID server. The Agrobacterium virulence gene database was selected, with the minimum gene length coverage set to 80% and maximum allowed sequence divergence set to 20%. The time for each Vir-Search analysis ranged from 2 to 5 min. Results suggested that the genome sequences for nearly all of the Agrobacterium isolates had homologs of virulence genes demonstrably necessary for pathogenicity by Agrobacterium, while the genome sequences for the isolates AC27/96, AC44/96, and 14-2641 did not (Fig. 3B, data for isolate 13-2099-1-2 shown). Contrary to the results from molecular diagnostics tests, the reads from isolate 13-626 failed to align to any virulence genes except for two (nocM, nocP) involved in nopaline transport. This isolate had the fewest number of useable sequencing reads and the highest number of contigs compared to the others, and results could have been a consequence of a poor assembly of the Ti plasmid.

Indeed, the qualities of the 14 assemblies were highly variable, likely reflecting the multi-partite structure of the agrobacterial genomes, presence of a linear replicon, and/or variation in depth of sequencing. We therefore used SPAdes v.3.6.2 to de novo assemble each of the genome sequences, with the exception of isolate 14-2641 (Bankevich et al., 2012). The total sizes of the assemblies were similar to those generated using Velvet and the qualities of the assemblies were high. But assemblies generated using SPAdes had proliferations in errors with palindromic sequence that appeared to be unique to isolates expected to have linear replicons. We informed the developers of the SPAdes software who immediately resolved the issue in SPAdes 3.7.0. Inspection of the summary statistics of the assemblies derived using this latest version of SPAdes suggested that relative to Velvet-based assemblies, there were improvements to all assemblies, with the most dramatic to those with the lowest read coverage (Table S1, Fig. S2). To further verify the quality of assemblies generated using SPAdes 3.7.0, we used Mauve to align Velvet and SPAdes assembled genome sequences of isolate 13-626 to the finished genome sequence of the reference sequence of A. radiobacter K84 (Darling et al., 2004; Slater et al., 2009). The SPAdes-based assembly was superior in being less fragmented and we were able to elevate the quality of the assembly from “unusable” to “high quality” (Figs. S2 and S3). Therefore, there is greater confidence in concluding that isolate 13-626 lacks the vir genes and T-DNA sequence. It does however have an ∼200 kb plasmid sequence which encodes nocM and nocP; this contig also encodes sequences common to replication origins of plasmids. We therefore suggest that because an isolate from the same pear gall sample originally tested positive for virD2, we mistakenly sequenced a non-pathogenic isolate.

Figure 4 Maximum likelihood tree based on vertically inherited polymorphic sites core to 20 Rhodococcus isolates.

WGS Pipeline was used to automate the processing of paired end short reads from 20 previously sequenced Rhodococcus isolates, and generate a maximum likelihood unrooted tree. Sequencing reads were aligned, using R. fascians strain A44a as a reference. SNPs potentially acquired via recombination were removed. The tree is midpoint-rooted. Scale bar = 0.05 average substitutions per site; non-parametric bootstrap support as percentages are indicated for each node. Major clades and sub-clades are labeled in a manner consistent with previous labels.

To validate the WGS Pipeline tool of Gall-ID, Illumina paired end read sequences derived from previously generated genome sequences of 20 Rhodococcus isolates were used to construct a phylogeny based on SNPs (Creason et al., 2014a). Using default parameters, the entire process, from piling up reads to generating the final phylogenetic tree, took 16 h (Table 1). A total of 855,355 sites (out of a total of 5,947,114 sites in the A44a reference sequence) were shared in at least 18 of the 20 Rhodococcus genome sequences. Of the shared sites, 177,961 sites were polymorphic, of which 3,142 were removed because they were identified as potentially acquired by recombination. The final core SNP alignment was therefore represented by 174,819 polymorphic sites and used to construct a maximum likelihood tree (Fig. 4). Most of the nodes were well supported, with all exceeding 68% bootstrap support and most having 100% support. The topology of the tree was consistent with that derived from a multi-gene phylogeny (Creason et al., 2014a). As previously reported, the 20 isolates formed two well-supported and distinct clades, and could explain the relatively low number of shared SNPs. The substructure that was previously observed in clade I was also evident in the ML tree based on core SNPs. The one noticeable difference between the trees was that the tips of the tree based on the core SNPs had substantially more resolution, and in particular, revealed a greater genetic distance between isolates A76 and 05-339-1, than previously appreciated based on the multi-gene phylogeny.

The amount of time to run Auto ANI was determined by comparing genome assemblies of the same 20 Rhodococcus isolates. The entire process was completed in five hours.

Conclusions

Gall-ID provides simplified and straightforward methods to rapidly and efficiently characterize gall-causing pathogenic bacterial isolates using Sanger sequencing or Illumina sequencing. Though Gall-ID was developed with a particular focus on these types of bacteria, it can be used for some of the more common and important agricultural bacterial pathogens. Additionally, the downloadable tools can be used for any taxa of bacteria, regardless of whether or not they are pathogens.

Supplemental Information

Table S1 Comparison of Velvet to SPAdes assemblies of 14 isolates associated with crown gall

Click here for additional data file.

Figure S1 Validation of the Phytopath-type tool

The 16S rDNA sequence from isolate originally labeled as Agrobacterium isolate 14-2641 was analyzed using the Phytopath-type tool. The isolate is labeled in red, as “query_isolate”; inset shows the clade that circumscribes the isolate.

Click here for additional data file.

Figure S2 The SPAdes assembler produces high quality assemblies

Genome sequences assembled using Velvet v. 1.2.10 (blue) and SPAdes v. 3.7.0 (orange) were compared based on scaffold N50 and number of scaffolds greater than 1 kb in size.

Click here for additional data file.

Figure S3 Whole genome alignment of 13-626 assemblies to a close reference sequence indicates collinearity of genomes

The A. radiobacter K84 genome sequence (top) and assemblies of 13-626 generated using SPAdes v. 3.7.0 (middle; 47 scaffolds) and Velvet (bottom; 527 scaffolds) were aligned using Mauve. Shared locally collinear blocks (LCBs) between the reference and the two assemblies are color-coded and connected by color-coded lines. Scaffold and replicon ends are depicted as vertical red lines.

Click here for additional data file.

We thank Melodie Putnam for providing the 14 bacterial isolates and for critical reading of the manuscript. We thank Dr. Pankaj Jaiswal for organizing and inviting us to participate in the STEM DNA Biology and Bioinformatics summer camps (Oregon State University). Camp participants Ana Bechtel, Mason Hall, Reagan Hunt, Pranav Kolluri, Benjamin Phelps, Joshua Phelps, Aravind Sriram, Megan Thorpe, and eight others, prepared genomic DNA and libraries for whole genome sequencing and analyzed the data. We thank Charlie DuBois of Illumina for providing kits for library preparation as well as sequencing, and Mark Dasenko, Matthew Peterson, and Chris Sullivan of the Center for Genome Research and Biocomputing for sequencing, data processing, and computing services. Finally, we thank the Department of Botany and Plant Pathology for supporting the computational infrastructure. Any opinion, findings, and conclusions or recommendations expressed in this material are those of the authors(s) and do not necessarily reflect the views of the US Department of Agriculture or National Science Foundation.

Additional Information and Declarations

Competing Interests

Author Contributions

DNA Deposition

Data Availability

JH Chang and NJ Grünwald are Academic Editors for PeerJ.

Edward W. Davis II and Alexandra J. Weisberg performed the experiments, analyzed the data, contributed reagents/materials/analysis tools, wrote the paper, prepared figures and/or tables, reviewed drafts of the paper.

Javier F. Tabima performed the experiments, analyzed the data, contributed reagents/materials/analysis tools.

Niklaus J. Grunwald conceived and designed the experiments, wrote the paper, reviewed drafts of the paper.

Jeff H. Chang conceived and designed the experiments, wrote the paper, prepared figures and/or tables, reviewed drafts of the paper.

The following information was supplied regarding the deposition of DNA sequences:

BioProject number PRJNA319063.

The following information was supplied regarding data availability:

All scripts are available from: http://gall-id.cgrb.oregonstate.edu/.

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
