# Peer review of "Gall-ID: tools for genotyping gall-causing phytopathogenic bacteria"

_PeerJ, doi:10.7717/peerj.2222_

## Round 0.1 · original submission · Minor Revisions

Please see the constructive comments from our three reviewers, below. Please also be sure to update the "demo" for users to follow easily.

Reviewer 1 ·

Basic reporting

The authors of the paper present a web based platform Gall_D, that use 16S rDNA sequence, multilocus sequence and whole genome analysis. They mainly target certain gall forming bacteria that causes external growths in plants. They also provide certain downloadable software pipelines that can be used for for further analysis.
The basic reporting and background for the paper is adequate, although I would like the authors to clarify certain issues with the paper.

Experimental design

Please see General Comments for the Author section.

Validity of the findings

Please see General Comments for the Author section.

Additional comments

1, The authors need to include the role of phylogeny, how the pathogen can evolve through time, integrated genes or genomic islands from closely related organisms? The bacterial genome evolve very dynamically.
2. I want the authors to clarify the affect of the antibiotic resistant genes, genome mobility, CRISPRs, transposable elements on the virulence of this bacteria.
3. Genome sizes are more conserved at all tested taxonomic levels than 16S rRNA copy numbers. Only a minority of bacterial genomes harbors identical 16S rRNA gene copies, and sequence diversity increases with increasing copy numbers. How do the authors account for such diversity?
4. "(MLSA) leverages the phylogenetic signal from four to ten genes" are these the housekeeping genes/ conserved genes ? Why is the set limited to 4-10, what is the rate of conservation of these genes.
5. Line 67: "phylogenetic analyses can be removed from studies to allow for robust analyses." I am a bit confused about this statement, can the authors please explain this in a better way?
6. Line 68: Are the authors aware of denovo assembly?, the later being more powerful as it takes into account structural changes in the genome, as bacterial genomes are highly mosaic, there is no dependence on any reference genome. Although sometime it is better to use a combination of de novo assembly and genome-guided assembly.
7. Line 77: I would like to stress on the authors need to shed some light on the genomic islands, pathogenicity islands that can be acquired by the bacterial genome? The virulence of the bacteria as stated in this paper is mainly caused by horizontal gene transfer, providing a brief background on that would help the readers.
8. "A Ti plasmid imparts, " change to "A Ti (tumor inducing plasmid) plasmid imparts", how big is this plasmid and which region does this gets integrated to. Please consider including certain important information about this plasmid, At least 25 vir genes on the Ti plasmid are necessary for tumor induction.
9. Line 164: However, one fact that has been overlooked is that multiple copies of this gene are often present in a given bacterium. These intragenomic copies can differ in sequence, leading to identification of multiple ribotypes for a single organism. Is there a way the tool takes into account multiple copies of the gene?
10. When taking into consideration sequence reads, the authors mention the reads need to be filtered, both quality filtering and removal of PCR and sequencing adapaters, so my question would be how important is the quality of the reads for the tools or does the tool do some sort of sanity check before running any analysis?
11. Line 290: Can you please provide the depth of coverage, read length or number of reads, such statistics are necessary to evaluate the quality of the sequences needed for such analysis?
12. Line 305: Denovo assemblers perform really poor with repeat regions, did the authors take into account such issue?
13. A detailed list of parameters for running Velveth and Spades would be very helpful
14. Line 482: Please see the bbmap software package, easy to use and really fast (https://sourceforge.net/projects/bbmap/), uses bbduk to do quality trimming and adapter removal.

Reviewer 2 ·

Basic reporting

The paper adhere to peerj policies, figures are relevant, source code and raw data have been provided.

Experimental design

No Comments

Validity of the findings

No Comments

Additional comments

General comments:
The paper presents a web-based platform that uses DNA sequence information from 16S rDNA, multilocus sequence analysis and whole genome sequences to group disease-associated bacteria to their taxonomic units.
The authors describe in details the case studies, the platform and all links and source code is shared on a public website.
While the paper is well written and presenting an interesting topic, the author may consider the comments below to improve the presentation of the tool and paper.

- Even if specific methods for parts of the pipelines are mentioned in the "introduction" section of the paper, the authors do mention or compare their new pipelines with existing ones, used by other groups in similar settings. How does this effort compare to others? Highlighting the differences in terms of performance, accessibility, ease of use or ease of customization may encourage users to adopt this platform.
- It is a good idea to have a "Demo" dataset for people to have a taste of the overall workflow and results. However it would be good to give a general estimation of the time it takes to process the demo file. Since that is a fixed file, it would be helpful to show a little note that says "processing this will take 2 minutes" or whatever the actual time is.
- After generating the tree for the "Demo", I still see the text "Please wait while the tree is generated" on the page. This is kind of confusing, because the tree is right below. Authors may need to fix this issue.
- The authors mention that the databases used in the pipeline are manually curated. There is no mention about updates and maintenance of this information, support for troubleshooting of the pipeline or any other user related query. Authors may need to consider adding this information on the main website.
- The generated tree for the "Demo" is very hard to read. Are there parameters that can be changed by the user to make it more readable? If yes, it may be worth to add a link to the document/instructions to do that. If no, it may be worth to add it, as a dynamic exploration of the data from the website could be of value.

Minor issues:
- (abstract) "have contributed to increasing the" should be "have contributed to increase the"
- In the tab "Auto ANI: Average Nucleotide Identity Analysis" the "Citations" section is empty. The authors may want to update this information.
- Clicking on the different tabs of the website result in a pretty long waiting time before anything appears on the page. I assume that the lag is due to the long loading time of the rshiny frame. However it may be slightly confusing for a user to just see a blank page for 5 seconds. I tested this with different browsers and the speed seems to be also affected by cache and other browser settings. The authors may think to add a "Loading" kind of image/text in place of the rshiny frame while that is loading.

Reviewer 3 ·

Basic reporting

The English written is good and the structure indeed followed the PeerJ sections requirements. But it is not easy to quickly get an overview of the manuscript, better to insert sub-section for reader's convenience.
Generally the manuscript has more than 5 pages in introduction section to introduce several "Gall bacteria" which is too much, especially for a tool description paper. It then illustrated all the function modules of the web tool Gall-ID, and their corresponding validation one-by-one all in results and discussion section, people would easily got lost in such overwhelming info without any sub-section or outline.

Experimental design

This manuscript described a web tool Gall-ID, which has, "Gall Isolate Typing", "Phytopath-Type", "Vir-Search", module function/tools in separated TAB, it also offers WGS Pipeline and manually curated databases to download. Descriptions of these modules, materials, methods, packages and instructions are in details and easy to follow up, the web tool itself is straight forward to try.
I have 2 questions below
1. A basic question to the Design is, what is the advantage of these molecular well tool to the proven methods? ("Proven methods for identification have been developed based on discriminative phenotypic and genotypic characteristics, including presence of antigens, differences in metabolism, or fatty acid methyl esters, and assaying based on polymorphic nucleotide sequences .") I noticed the validation were conducted on 14 isolates, how about the performance of these proven methods on them?

2. "Users must first select the appropriate taxonomic group, Agrobacterium, Pseudomonas, Pantoea, or Rhodococcus for comparison." What if the user has no idea about this or what if the isolate belongs to non of these 4 groups?

Validity of the findings

For timing issue, I just tried several module functions with "DEMO" dataset but not WGS or Vir-Search. For example 16S demo for Agro-type, it takes about tens of seconds to process and finally generated the distance tree with bootstrap and a minimum spanning network.

Not sure the size and time for real data, but I would assume that would be larger/longer than DEMO. It is better to discuss that in the paper.

Also I can not understand the final generated spanning network, see attached figure. What are these pie chart in the network around each node? Why some pie chart sub-sections have the same color? what does this mean? Please illustrate in details in paper.

Annotated reviews are not available for download in order to protect the identity of reviewers who chose to remain anonymous.

---

## Round 0.2 · accepted · Accept

Please address a few minor issues from reviewers while in production.

Reviewer 1 ·

Basic reporting

No Comments

Experimental design

No Comments

Validity of the findings

No Comments

Additional comments

The authors have accurately answered all my queries and concerns.

Reviewer 2 ·

Basic reporting

The paper adhere to peerj policies, figures are relevant, source code and raw data have been provided.

Experimental design

No Comments

Validity of the findings

No Comments

Additional comments

The authors addressed all my previous comments in a satisfactory way.

Few minor issues that they may want to fix:
- a "loading" image has been included in the website to account for the delay in displaying the rshiny frame. This image seems to work in Firefox but not in Explorer
- about my comment on the readability of the generated tree, in their rebuttal authors say "from the website, users can manually zoom in on branches of interest". Is the zoom triggered by a keyboard combination? Mouse? This information could be added to the website so that users know what to click/do if they want to zoom in/out.

Reviewer 3 ·

Basic reporting

No Further Comments

Experimental design

No Further Comments

Validity of the findings

No Further Comments